# Textless Low-Resource Speech-to-Speech Translation With Unit Language Models

## Abstract

Existing speech-to-speech translation models fall into two camps: textless models trained with hundreds of hours of parallel speech data or unsupervised models that leverage text as an intermediate step. Both approaches limit building speech-to-speech translation models for a wide range of languages, as they exclude languages that are primarily spoken and language pairs that lack large-scale parallel speech data. We present a new framework for training textless low-resource speech-to-speech translation (S2ST) systems that only need dozens of hours of parallel speech data. We reformulate S2ST as a unit-to-unit seq2seq translation task, and start by pretraining a model on large-scale monolingual speech data. Then, we finetune it with a small amount of parallel speech data ($20 - 60$ hours). Lastly, we improve model performance through an unsupervised backtranslation objective. We train and evaluate our models for English-to-German, German-to-English and Marathi-to-English translation on three different domains (European Parliament, Common Voice, and All India Radio) with single-speaker synthesized speech data. Evaluated using the ASR-BLEU metric, our models achieve reasonable performance on all three domains, with some being within 1-2 points of our supervised topline.

## 1 Introduction

The speech-to-speech translation (S2ST) task involves translating input speech in the source language to speech in the target language. In many ways, S2ST represents the "holy grail" of translation as it enables natural, real-time, spoken communication. S2ST has a rich history, from cascaded systems combining Automatic Speech Recognition (ASR), Machine Translation (MT), and Text To Speech (TTS) technologies (Nakamura et al., 2006) to recently proposed neural end-to-end systems (Lee et al., 2022a; Seamless Communication et al., 2023) that directly map from input source language speech to output target language speech. S2ST systems (Jia et al., 2019; Lee et al., 2022a;b; Jia et al., 2021; Duquenne et al., 2022; Seamless Communication et al., 2023) have benefited from model and data scaling, leveraging increasing amounts of parallel speech and/or text data across languages. Yet, this is feasible only for a fraction of the world's 7000 languages (Lewis et al., 2016); the majority of world languages have low-resource or no parallel translation data available (Haddow et al., 2022). Furthermore, thousands of languages are primarily spoken without standardized writing systems (about 3000 languages in Ethnologue (Lewis et al., 2016) have no reported writing system), necessitating textless language processing.

Recent work on textless speech translation (Lee et al., 2022b; Kim et al., 2023) requires large amounts of parallel cross-lingual speech data, making it difficult to adapt for low-resource speech translation. On the other hand, some other papers have proposed approaches for training S2ST models that do not need any parallel speech data at all; however, these approaches either train cascaded models that have intermediate text outputs (Wang et al., 2022a; Fu et al., 2023) or use text supervision during training (Nachmani et al., 2023). As a result, these are difficult to adapt for speech translation on languages (spoken, with non-standard orthographies or poor ASR) that would benefit from purely textless approaches.

We propose a learning framework that requires a much more modest amount (dozens of hours) of parallel speech data to train a textless speech-to-speech translation model. We begin by pretraining an encoder-decoder

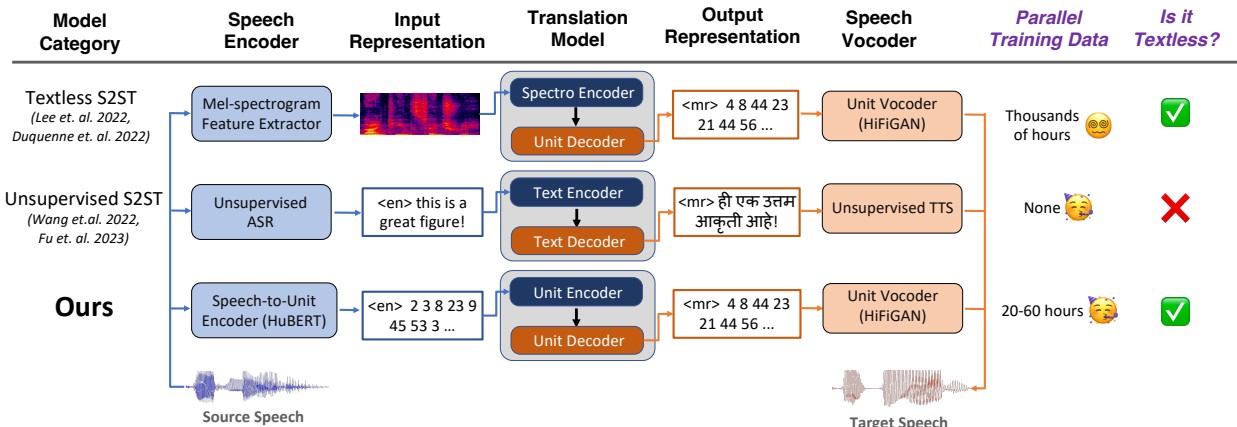

Figure 1: Overview of speech-to-speech translation systems. We compare our formulation to two relevant lines of work. We present the first textless speech-to-speech system that does not require a large-scale parallel training dataset.

language model over self-supervised speech units using non-parallel speech corpora, followed by finetuning it for S2ST by finetuning on a low-resource parallel S2ST corpus and finally performing unsupervised backtranslation to further improve performance. We achieve this by reformulating S2ST as a unit-to-unit machine translation problem. Figure 1 illustrates our method, comparing it to previous work. Modelling real speech data with speech unit sequences poses challenges, such as inherent unit sequence noise and ambiguity, that are orthogonal to our research questions. Thus, for simplicity, we use single-speaker synthesized speech data to train and evaluate our models, following early S2ST work (Jia et al., 2019).

We train two English ↔ German S2ST models in the European Parliament (Iranzo-Sánchez et al., 2019) and Common Voice (Ardila et al., 2020) domains and two English ↔ Marathi S2ST models in the European Parliament (Iranzo-Sánchez et al., 2019) and All India Radio (Bhogale et al., 2022) domains, and evaluate the en→de, de→en and mr→en translation directions. We find that with just 20 hrs of parallel en→de and de→en data and 60 hrs of parallel en→mr and mr→en data, our models achievable reasonable performance on all three domains, obtaining ASR-BLEUs of 10.0 (de→en), 8.3 (en→de) and 9.2 (mr→en) for the European Parliament domain, 7.7 (de→en) for the Common Voice domain, and 10.0 (mr→en) for the All India Radio domain. Our results are within 1-2 ASR-BLEU of our high-resource supervised topline for the European Parliament domain for the de→en and mr→en language pairs. We will release code and model weights at the time of publication.

## 2 Methods

We represent the input and output speech utterances as discrete unit sequences and train a unit-based encoder-decoder model for the speech-to-speech translation task. Therefore, our pipeline consists of a speech-to-unit encoder (S2U), a unit encoder-decoder (U2U) and a unit-to-speech vocoder (U2S). Of these, S2U and U2S are essentially speech-unit interfaces; we base these largely on prior work (Hsu et al., 2021; Polyak et al., 2021). Our main contribution is the middle unit-based encoder-decoder model (U2U) that is trained for S2ST using our three-step Pretrain-Finetune-Backtranslate approach illustrated in Figure 2. We now describe each of these components below.

### 2.1 Speech-to-unit Encoder (S2U)

We first describe the model we use to map speech waveforms into a sequence of discrete unit representations. Past work (Hsu et al., 2021; Chung et al., 2021) has explored learning self-supervised discrete representations of speech. The learned discrete representations, or units, preserve much of the information contained in the

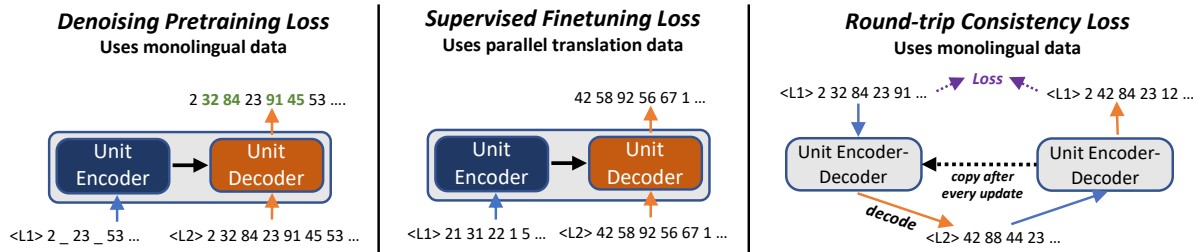

Figure 2: Training a unit-based encoder-decoder model for speech-to-speech translation. The first **Pretrain** step trains on large-scale monolingual speech data using a denoising pretraining loss. The second **Finetune** step trains on low-resource (20-60 hours) of parallel speech-speech translation data using a supervised finetuning loss. The third **Backtranslate** step trains using a combination of a round-trip consistency loss (on monolingual data) and the supervised finetuning loss (on parallel data) used in the second step.

original input signal (Pasad et al., 2021), including phonemes, word identity, speaker identity, and so forth. Critically, text transcriptions or other annotations of the speech are not necessary to discover these units. It has recently become popular in the research community to train autoregressive language models (Lakhotia et al., 2021; Borsos et al., 2022) on these unit representations, enabling NLP tasks to be performed on spoken language without the need to first transcribe speech waveforms into text.

We base our speech-to-unit encoder on the pre-trained HuBERT (Hsu et al., 2021) base model. As proposed by HuBERT (Hsu et al., 2021), we train a k-means clustering model over HuBERT embeddings at an intermediate layer, choosing the layer index on the basis of the units' PNMI score, a phone-unit mutual information metric. We train a shared English-German k-means model and a separate Marathi k-means model, our best configuration. To convert a speech waveform to a unit sequence, we pass it through HuBERT, extract embeddings at an intermediate layer, use the k-means clustering model to map each timestep's embedding to its nearest cluster center, and apply run-length encoding (collapsing consecutive equal units into one) as in prior work (Lee et al., 2022b). A unit sequence is thus a sequence of integers corresponding to indices of mapped clusters. We also experimented with other models, XLSR (Conneau et al., 2020) and Indic-wav2vec (Javed et al., 2021), but decided to use HuBERT on the basis of its units' high PNMI score. We describe training the clustering model and the evaluation of the speech-to-unit encoder in Section 4.1.

## 2.2 Unit Encoder-Decoder (U2U)

We train our unit-based encoder-decoder model to perform S2ST using a three-step Pretrain-Finetune-Backtranslate approach visualized in Figure 2. We describe each step in this section, and provide implementation details in Section 4.2.

**Pretrain** We initialize the model with mBART-50 (Liu et al., 2020) (a text encoder-decoder model), reinitialize the input and output embedding layers for our new unit vocabulary, and pretrain using their original denoising objective. While we initialize with mBART-50, we feed it unit sequences, which do not exist in the text token space. However, since unit sequences can be treated as text sequences, just with a different vocabulary, we can easily adapt the training pipeline to train on unit sequences rather than text sequences. Given a unit sequence dataset $\mathcal{D}$ and a noising function $g(\cdot)$ (we use one that samples contiguous spans and masks them until a fixed ratio of tokens are masked), the decoder is trained to generate the original sequence $X$ given encoder input $g(X)$, optimizing model weights $\theta$ as $\arg\min_\theta \sum_{X \in \mathcal{D}} -\log \Pr(X|g(X); \theta)$.

We train two bilingual unit LMs, one for English-German, and one for English-Marathi. They are trained on unit sequences, derived from monolingual speech corpora in the three languages, generated by the respective S2U encoder (shared for English-German and separate for Marathi). We train one Sentencepiece (Kudo & Richardson, 2018) BPE tokenizer per LM to create the vocabulary.

**Finetune**  We perform supervised training on the pretrained unit LM using a small parallel S2ST corpus, where the input is a spoken utterance in the source language, and the target is a translated version spoken in the target language. During this finetuning process, we use the standard cross-entropy loss of the decoder generating the target unit sequence, when the ground truth source unit sequence is provided to the encoder.

**Backtranslate**  Finally, we perform unsupervised backtranslation (Lample et al., 2018) on our finetuned model. We follow the standard recipes used in unsupervised text backtranslation, with minor modifications to stabilize training in the speech domain. We briefly describe the procedure: unsupervised backtranslation trains the model to reconstruct a unit sequence from a model-generated synthetic translation of the same unit sequence using a round-trip translation consistency loss (visualized in Figure 2). For every training step, denoting the model as $\mathcal{M}$,

1. Get a batch of utterances in one language, $B_1$, and a batch of utterances in another language, $B_2$.
2. Use $\mathcal{M}$ to translate $B_1$ to translations $B_1'$, and $B_2$ to translations $B_2'$; this step is inference only and no gradient updates occur.
3. Given $B_1', B_2'$ as input respectively, compute the decoder cross-entropy loss for the model $\mathcal{M}$ to reconstruct the original utterances $B_1, B_2$. Using this loss, perform a gradient update on $\mathcal{M}$'s parameters.

The above corresponds to online backtranslation, where the 'forward' model (generating the synthetic translation) is the same as the 'backward' model (used to compute the cross-entropy loss). We also explored offline backtranslation, which updates the forward model every epoch, but did not see much difference in performance. Unlike in unsupervised text backtranslation, the training was unstable in both settings. To resolve this, we mix in some supervised data (used in the finetuning step) with online backtranslation during this last stage, which stabilizes learning and shows gains.

### 2.3  Unit-to-speech Vocoder (U2S)

We adapt prior work (Polyak et al., 2021)[1] on speech resynthesis from discrete units to build our unit-to-speech vocoder. Given a dataset consisting of speech waveforms and their corresponding unit sequences generated by the S2U encoder, the model trains two submodules; a duration prediction module and a HiFi-GAN (Kong et al., 2020) that converts unit sequences back to speech waveforms. The duration predictor is a two-layer CNN that takes a run-length-encoded unit sequence as an input, predicts the duration of each unit, and repeats each unit to match its predicted duration. The HiFi-GAN generator consists of a sequence of transposed CNNs that take full unit sequences as input and sequentially upsample the sequence to obtain speech waveforms as output. The HiFi-GAN is trained as a GAN with this generator and a set of CNN discriminators. We train separate U2S vocoders for each language (English, German, Marathi).

## 3  Experimental Setup

### 3.1  Datasets

Table 1 summarizes datasets used in our work. For each language pair, we train models on different domains. Durations reported for parallel translation datasets correspond to durations of the source speech. More dataset details are in Table 4 of Appendix A.

**English-German**  For pretraining, we use the union of the transcribed set of Voxpopuli (Wang et al., 2021) and randomly-sampled subsets of the Europarl v3 (Koehn, 2005) train set that we call Europarl-small and Europarl-mid (refer to Table 4 of Appendix A for statistics), collected from European Parliament recordings. For finetuning, we use two datasets: (1) randomly-sampled 20-hr (10-hr per translation direction i.e. en→de and de→en) subset of the Europarl-ST (Iranzo-Sánchez et al., 2019) train set and (2) randomly-sampled 20-hr (10-hr per translation direction) subset of the CVSS (Jia et al., 2022) train set. For the last backtranslation

---

[1] https://github.com/facebookresearch/speech-resynthesis/tree/main/examples/speech_to_speech_translation

| Model Name | Languages | Pretrain | Finetune | Backtranslate | Evaluation |
|---|---|---|---|---|---|
| $M\mathtt{de}^{\mathtt{EP}}$ $M\mathtt{de}^{\mathtt{CV}}$ | de,en | VP (777h) + EP (5381h) | EP-ST (20h) CVSS (20h) | VP (777h) CV (382h) | EP-ST (9h) en↔de CVSS (15h) de→en |
| $M\mathtt{mr}^{\mathtt{EP}}$ $M\mathtt{mr}^{\mathtt{Shr}}$ | mr,en | VP (529h) + Shr (1000h) | s-Ep-ST (60hr) s-Shr-ST (60hr) | VP (529h) + Shr (1000h) | s-Ep-ST (9h) mr→en s-Shr-ST (10h) mr→en |

Table 1: Model configurations. For each dataset, we mark their duration in parentheses. Abbreviations: VP = Voxpopuli, EP = Europarl, EP-ST = Europarl-ST, CV = CommonVoice, Shr = Shrutilipi, S-Ep-ST = synth-Europarl-ST, S-Shr-ST = synth-Shrutilipi-ST.

step, we use Voxpopuli and Common Voice 4 (Ardila et al., 2020) data for the round-trip consistency loss. Common Voice and CVSS are collected using the Mozilla Common Voice project and consist of recordings of crowd-sourced workers reading out sentences primarily derived from Wikipedia; thus they do not belong to the European Parliament domain. For evaluation, we use Europarl-ST (Iranzo-Sánchez et al., 2019) (for both de→en and en→de) and CVSS (Jia et al., 2022) (for de→en) test sets.

**English-Marathi**  For pretraining, we use the union of the Shrutilipi (Bhogale et al., 2022) transcribed Marathi dataset, collected from All India Radio broadcasts and the English transcribed train set of Voxpopuli. We were unable to find domain-matched speech translation datasets for Marathi-English. Thus, we synthetically generate parallel datasets by translating the source language utterance to target language utterance using the Google Translate API[2]. An author of this paper, who speaks both Marathi and English, manually checked a few utterances and found the translations to be of high quality. We construct two such datasets, each consisting of train and test sets: (1) Synth-Europarl-ST: translating the English side of the English-German Europarl-ST train and test sets to Marathi. (2) synth-Shrutilipi-ST: translating 100-hr and 10-hr subsets of the Shrutilipi dataset to English, creating a train and test set respectively.

For finetuning, we randomly sampled 60-hr (30-hr per translation direction) subsets of the train sets of these two datasets. We empirically found that we need more data in English-Marathi compared to English-German, which we hypothesize is due to greater language and domain dissimilarities. For the backtranslation step, we use the union of Voxpopuli and Shrutilipi datasets for the round-trip consistency loss. For evaluation, we use the test sets of these Synth-Europarl-ST (where Marathi is translated from English), and synth-Shrutilipi-ST datasets, (where English is translated from Marathi). We only evaluate the mr→en translation direction for both. None of the targets in the test sets of either dataset have been seen during pretraining, making them suitable for use.

### 3.2   Model Configurations

Table 1 describes training and evaluation datasets for each of our four models. $M\mathtt{de}^{\mathtt{EP}}$ is trained and evaluated entirely within the European Parliament domain: it is pretrained on the union of Voxpopuli and Europarl v3, finetuned on Europarl-ST, backtranslated with Voxpopuli, and evaluated on Europarl-ST. $M\mathtt{de}^{\mathtt{CV}}$ uses the same pretraining, but is finetuned on CVSS, backtranslated with Common Voice 4.0, and evaluated on CVSS. Common Voice and CVSS consist of crowd-sourced speech recordings reading aloud sentences primarily derived from Wikipedia, which differ from the European Parliament domain. $M\mathtt{mr}^{\mathtt{EP}}$ and $M\mathtt{mr}^{\mathtt{Shr}}$ are both pretrained and backtranslated with the union of Voxpopuli and Shrutilipi i.e. English European Parliament data and Marathi All India Radio data. $M\mathtt{mr}^{\mathtt{EP}}$ is finetuned and evaluated on the European Parliament domain using synth-Europarl-ST while $M\mathtt{mr}^{\mathtt{Shr}}$ is finetuned and evaluated on the All India Radio domain using synth-Shrutilipi-ST. All four models are thus finetuned and evaluated with the same dataset's train and test sets.

---

[2]https://cloud.google.com/translate/docs/advanced/batch-translation

### 3.3 Generating Synthetic Speech Data

We use single-speaker synthesized speech data for both training and evaluation, following early S2ST work (Jia et al., 2019). All of our training datasets have ground truth transcripts; thus, we use TTS models to regenerate the speech from these transcripts and use the synthesized speech in our experiments. To generate synthetic speech data for English and German, we use Coqui-AI's TTS software.[3] These are VITS (Kim et al., 2021) models, a conditional VAE trained with an adversarial learning objective, trained on LJSpeech (Ito & Johnson, 2017) and Thorsten (Müller & Kreutz), each of which contain around 24 hrs of clean read speech. We use IndicTTS (Kumar et al., 2023) model for Marathi; this is a FastPitch (Łańcucki, 2021) model trained on the IndicTTS Database (Baby et al., 2016) that contains around 3 hrs of clean read speech.

## 4 Model Implementation

### 4.1 Speech-to-Unit Encoder (S2U)

We build our speech-to-unit encoder using k-means clustering over the embeddings produced by a self-supervised speech encoder model. We decide (a) which speech encoder model to use, (b) whether to learn separate per-language k-means models or a joint k-means model and (c) which encoder layer take embeddings from. We measure the average Pointwise Normalized Mutual Information (PNMI) between unit sequences and phoneme sequences extracted from the same datasets, following Hsu et al. (2021), choosing unit sequence that yields higher PNMI. We compare HuBERT (Hsu et al., 2021) and XLSR (Conneau et al., 2020) for English and German, and HuBERT and Indic-wav2vec (Javed et al., 2021) for Marathi for (a); we try all combinations for (b); and we try several layers for (c). To train the k-means models, we use $\approx 50$ hrs of raw speech data from each language, obtained from a random subset of Librispeech (Panayotov et al., 2015) for English, Multilingual Librispeech (Pratap et al., 2020) for German, and Shrutilipi (Bhogale et al., 2022) for Marathi. Our best configuration uses a Marathi k-means model (with 100 clusters) and a shared English-German k-means model (with 200 clusters). We find that this works better than training three individual models or a single model, which we hypothesize is due to similarity between English and German. For German and English, we use the 6th layer of HuBERT, while for Marathi we use the 8th layer. The details can be found in Appendix C.

### 4.2 Unit Encoder-Decoder (U2U)

**Preprocessing** We train one Sentencepiece BPE tokenizer per LM on the speech units with a 10000-size vocab, using Voxpopuli for the English-German LM and the union of Voxpopuli and Shrutilipi for the English-Marathi LM.

**Pretrain** Both LMs are initialized with the `mbart-large-50` (Liu et al., 2020) Huggingface checkpoint except the input and output embedding layers, which are reinitialized. The noising function $g$ is defined similarly to mBART; until the number of masked tokens reaches 35%, we sample span length $l$ from a Poisson distribution with mean $\lambda$ and replace a random contiguous unit sequence of length $l$ with a single MASK token. For English-German model, we pretrain it in several stages, increasing the task difficulty by masking longer spans in later stages. We first train on Voxpopuli for 900k updates with a Poisson lambda of 2. We then train on a combination of Voxpopuli and Europarl-small for 5400k; 2700k updates with Poisson lambda of 2 and 2700k updates with lambda of 8 (harder task due to longer spans). We finally train on a combination of Voxpopuli and Europarl-mid for 2700k updates. For English-Marathi, we only perform a single round, training on a combination of Voxpopuli and Shrutilipi with a Poission lambda of 2 for 900k updates.

For both LMs, we use an LR scheduler that starts with an LR of 1e-7, ramps up linearly to 1e-5, and then decays exponentially to 1e-6. We train on 4 GPUs. We use variably sized batches so that shorter sequences can be packed into larger batches; the total number of tokens in a batch is a maximum of 3125 tokens per language for English-German and 6250 tokens per language for English-Marathi, with equal amounts of tokens per language.

---

[3]We use the `en/ljspeech/vits` model for English and `de/thorsten/vits` model for German. `https://github.com/coqui-ai/TTS`)

**Finetune**  We use label smoothing, dropout of 0.2 and a learning rate of 3e-5. We train for 40 epochs with a total batch size of 3748 tokens on 4 GPUs. We finetune all parameters of the models except for $M\text{de}^{\text{EP}}$, for which we finetune only the last 5 layers of both encoder and decoder as it shows performance gains.

**Backtranslate**  When sampling translations during forward translation, we use nucleus sampling (Holtzman et al., 2019) with top-p value of 0.9 and the temperature of 0.5. We use label smoothing of 0.2, learning rate of 3e-5 and train for 3 epochs with a total batch size of 3748 tokens on 4 GPUs.

### 4.3  Unit-to-Speech Vocoder (U2S)

A separate vocoder is trained for each language, mapping from the unit vocabulary (size 200 for English-German, size 100 for Marathi) to speech clips at 16kHz. Using the unit sequences for the Voxpopuli (English and German) and Shrutilipi (Marathi) datasets, generated from our S2U encoder, we train vocoders to generate the speech from these unit sequences. We train across 4 GPUs with a learning rate of $2e - 4$ with a batch size of 128 (for en-de) and 240 (for mr) and train for 60k updates; other hyperparameters follow Polyak et al. (2021). As a sanity check, we evaluate S2U and U2S by computing the resynthesis WER, which measures how well passing a given speech signal through S2U and U2S preserves the content of the input speech signal. We find that our models perform comparably to previous models (Lee et al., 2022a). More details about this evaluation are in Appendix D.

## 5  Results

### 5.1  Evaluation Setup

We use the ASR-BLEU evaluation metric following prior work (Lee et al., 2022a;b): given a hypothesis speech translation and a ground truth text translation, we run ASR on the generated speech and compute the BLEU between the ASR transcript and the ground truth text translation with SacreBLEU's default parameters. We evaluate the de→en, en→de and mr→en language directions. We opted to not evaluate the en→mr direction due to poor Marathi ASR models that resulted in excessively noisy ASR-BLEU scores. We generate translations from our models using beam search decoding with a beam size of 10. When evaluating on Europarl-ST dataset, we use wav2vec2.0 based ASR models with greedy decoding (`facebook/wav2vec2-large-960h-lv60-self` and `jonatasgrosman/wav2vec2-xls-r-1b-german`) used by prior S2ST work on Europarl-ST (Duquenne et al. (2022); Wang et al. (2022b) and others). When evaluating on CVSS dataset, we use a medium-sized Whisper ASR model used by prior S2ST work on CVSS (Fu et al., 2023). When evaluating Marathi-English translation, we use the `facebook/wav2vec2-large-960h-lv60-self` ASR model.

### 5.2  Comparison Systems

Our results in Tables 2 and 3 compare several speech translation systems.

**Topline Models**  We compare our approach to existing models which use **more** resources:

- **Speech-to-text (S2T) models trained on large-scale parallel speech-text translation data**. ⓐ (Iranzo-Sánchez et al., 2019) is an ASR-MT cascade model whose MT component is trained on a large-scale text translation dataset OPUS (Tiedemann, 2012). ⓑ and ⓒ are Transformer-based models from Wang et al. (2021) trained on the union of Europarl-ST and CVSS (total duration 226h) with ⓒ being additionally trained on ≈300h of Voxpopuli aligned speech translation data.
- **Speech-to-speech translation (S2ST) models trained on large-scale parallel speech-text translation data**. ⓓ is the Translatotron 2 (Jia et al., 2021), a spectrogram-to-spectrogram encoder-synthesizer model trained with text supervision for the decoder with 120h of German-English data and about 360h of aligned data in 3 other X-to-English language pairs.
- **S2ST models trained without parallel data, but trained on large-scale monolingual text data.** ⓔ is a model by Fu et al. (2023) cascading an unsupervised ASR - unsupervised MT - unsupervised TTS pipeline.

- **Textless speech-to-speech translation (S2ST) models trained on large-scale parallel speech-speech translation data**. ⓕ is a bilingual S2ST model trained on a large, mined Speech-Matrix dataset ($\approx$ 2600 hrs of source speech for the en→de and the de→en directions combined) by Duquenne et al. (2022). ⓖ (Kim et al., 2023) is multilingual S2ST model trained on 650h of parallel aligned English-German Voxpopuli data, and about 12k hours of parallel aligned data in 18 other X-to-English language pairs. ⓗ and ⓞ present our pretrained unit LMs fine-tuned on large-scale data i.e. the Europarl-ST train set (110 hours), the CVSS train set (180 hours), the SYNTH-EUROPARL-ST train set (125h) and the SYNTH-SHRUTILIPI-ST train set (176h) using the same hyperparameters as our four low-resource models.

**Our Low-Resource Models** We train four models on different domains: $M\mathtt{de}^{\mathtt{EP}}, M\mathtt{de}^{\mathtt{CV}}, M\mathtt{mr}^{\mathtt{EP}}$ and $M\mathtt{mr}^{\mathtt{Shr}}$ as described in Section 3.2. We evaluate each model with its in-domain evaluation data, i.e., $M\mathtt{de}^{\mathtt{EP}}$ model on Europarl-ST, $M\mathtt{de}^{\mathtt{CV}}$ model on CVSS, $M\mathtt{mr}^{\mathtt{EP}}$ on SYNTH-EUROPARL-ST, and the $M\mathtt{mr}^{\mathtt{Shr}}$ model on SYNTH-SHRUTILIPI-ST. ⓘ and ⓟ report the model performance after our pretraining and finetuning steps. ⓙ and ⓠ report the model performance after performing backtranslation.

## 5.3 Main Results

We present our results for the English-German pair in Table 2 and the results for the English-Marathi pair in Table 3. Comparing the text-based S2T/S2ST topline models (ⓐ-ⓓ) with the textless S2ST topline models (ⓕ-ⓗ), we can see that the textless S2ST models, despite being trained with much more data in some cases, underperform the text-based S2T/S2ST models. This showcases the difficulty of learning a textless S2ST model. S2T models also do not suffer from ASR errors introduced at evaluation time, which is required for all other systems that produces speech. Our topline model ⓗ outperforms row ⓕ and row ⓖ for en→de translation despite using much less data, indicating the benefits of pretraining.

Now, we discuss our models trained on low-resource settings. We can see from rows ⓘ and ⓟ that our pretrained models, given only 20 hr of parallel data (for English-German) and 60 hr of parallel data (for English-Marathi), learn S2ST models with reasonable BLEU scores. Performing backtranslation consistently improves model performance, resulting in our best low-resource models in rows ⓙ and ⓠ. Our de→en Europarl-ST performance and the mr→en SYNTH-EUROPARL-ST performance is within 1-2 BLEU of our supervised toplines ⓗ and ⓞ despite being trained on much less data. However, our models underperform the textless high-resource (rows ⓕ and ⓖ) and text-based zero-resource (row ⓔ) S2ST models overall, leaving room for future work.

## 5.4 Ablations

We perform ablations for the $M\mathtt{de}^{\mathtt{EP}}$ model evaluated on the Europarl-ST test set to justify our modeling choices.

**Ablating pretraining** Our LM is initialized from the text mBART checkpoint, and then trained on a unit-based denoising objective. Without this pretraining (i.e., finetuning and backtranslating with the base mBART checkpoint), as seen in rows ⓚ and ⓛ, we obtain very low ASR-BLEUs less than 2 points. These results suggest that unit LM pretraining is essential in order to learn good S2ST systems in low-resource settings.

**Ablating finetuning** We train an unsupervised S2ST model, which is trained with a backtranslation round-trip consistency loss on top of the pretrained unit LM. The result, ⓜ, shows that this does not work, with near-zero BLEU scores. This suggest some amount of parallel speech is necessary.

**Ablating replay in backtranslation** We have already seen that adding backtranslation after finetuning boosts performance by 1-2 BLEU, demonstrated by comparing row ⓘ to ⓙ or row ⓟ to ⓠ. We replay the

---

[4]In addition to 120h of parallel German-English data, Translatotron 2 is trained on X-to-English translation data from 3 other languages, totalling $\approx$ 480 hours of parallel data.

[5]In addition to 650h of parallel German-English data, UTUT is trained on X-to-English translation data from 18 other languages, totalling $\approx$ 12000 hours of parallel data.

|  | | ASR-BLEU ↑ | | |
|  | | Europarl-ST | | CVSS |
| Model | Parallel #hrs | de→en | en→de | de→en |
| **Topline models** | | | | |
| *Text-based High-Resource S2T/S2ST models* | | | | |
| ⓐ Cascaded ASR-MT (Iranzo-Sánchez et al., 2019) | N/A | 21.3 | 22.4 | - |
| ⓑ E2E S2T (Wang et al., 2021) | 226h | 17.5 | - | - |
| ⓒ E2E S2T w/ Voxpop-Aligned (Wang et al., 2021) | ≈500h | 18.8 | - | - |
| ⓓ Translatotron 2 (Jia et al., 2021) | 120h [4] | - | - | 19.7 |
| *Text-based Zero-Resource S2ST* | | | | |
| ⓔ UASR → UMT → UTTS (Fu et al., 2023) | 0h | - | - | 14.7 |
| *Textless High-Resource S2ST* | | | | |
| ⓕ Bilingual S2S (Duquenne et al., 2022) | ≈2600h | 16.3 | 10.1 | - |
| ⓖ Multilingual UTUT (Kim et al., 2023) | 650h [5] | 15.8 | 9.8 | - |
| ⓗ Pretrain + Fully Finetune (Ours) | 110h\|180h | 12.0 | 13.4 | 13.6 |
| *Textless Low-Resource S2ST* | | | | |
| ⓘ Pretrain + Finetune (Ours) | 20h | 7.8 | 6.8 | 5.8 |
| ⓙ + Backtranslate (Ours) | 20h | 10.0 | 8.3 | 7.7 |
| **Ablations** | | | | |
| *Ablating Pretraining* | | | | |
| ⓚ Text mBART + Finetune | 20h | 1.0 | 0.3 | - |
| ⓛ + Backtranslate | 20h | 1.3 | 0.4 | - |
| *Ablating Finetuning* | | | | |
| ⓜ Pretrain + Backtranslate | 0h | 0.4 | 0.1 | - |
| *Ablating Backtranslation Replay* | | | | |
| ⓝ Pretrain + Finetune + Backtranslate w/o replay | 20h | 4.3 | 4.0 | - |

Table 2: English-German S2ST evaluation using the ASR-BLEU metric on Europarl-ST (Iranzo-Sánchez et al., 2019) and CVSS (Jia et al., 2022) test sets; higher is better. Topline models have either been trained on high-resource supervised datasets, or are not textless due to use of intermediate text generation; see Section 5 for discussions. The Parallel #hrs column denotes the number of hours of parallel translation training data. In ⓗ it denotes 110h of EP-ST data and 180h of CVSS data is used to train two separate topline models.

supervised low-resource finetuning data during backtranslation to stabilize training. We ablate training with this replay by running the backtranslation step with just the round-trip consistency loss. The result, row ⓝ, shows that the performance worsens compared to the initialization of row ⓘ. With replay, however, we get the results in row ⓙ, which improve upon the initialization.

# 6 Related Work

## 6.1 Speech-to-Speech Translation (S2ST)

While cascaded S2ST models (Nakamura et al., 2006; Wahlster, 2000) that generate intermediate text translations (either as an ASR-MT-TTS or an S2T-TTS cascade) have existed for a long time, end-to-end S2ST models can be traced back to Jia et al. (2019) who trained a model that directly translates source language speech waveforms to speech waveforms in the target language. While most S2ST systems directly predict speech waveforms at inference time, some S2ST models (Jia et al., 2019; 2021; Lee et al., 2022a; Inaguma et al., 2022) are text-based i.e. they opt to use textual supervision during training to stabilize system components or to obtain improved performance, while other S2ST models (Lee et al., 2022b; Li

| | | ASR-BLEU ↑ | |
| | | synth-EP-ST | synth-Shr-ST |
| Model | Parallel #hrs | mr→en | mr→en |
| --- | --- | --- | --- |
| **Topline models** | | | |
| *Textless High-Resource S2ST* | | | |
| ⓞ Pretrain + Finetune (Full) (Ours) | 125h\|176h | 10.9 | 17.8 |
| *Textless Low-Resource S2ST* | | | |
| ⓟ Pretrain + Finetune (Ours) | 60h | 8.3 | 9.6 |
| ⓠ + Backtranslation (Ours) | 60h | 9.2 | 10.0 |

Table 3: Marathi-English S2ST evaluation using the ASR-BLEU metric on our SYNTH-EUROPARL-ST and SYNTH-SHRUTILIPI-ST test sets; higher is better. Topline models have been trained on high-resource supervised datasets; see Section 5 for discussions. The Parallel #hrs column denotes the number of hours of parallel translation training data. In ⓞ it denotes 125h of SYNTH-EUROPARL-ST data and 176h of SYNTH-SHRUTILIPI-ST data is used to train two separate topline models.

et al., 2022; Kim et al., 2023; Zhu et al., 2023) are trained in a textless manner, representing speech using self-supervised speech units, potentially paving the way to extend S2ST technology to hundreds of languages that are primarily spoken or have very bad ASR systems. Most of these S2ST models, especially the textless ones, require large training datasets of parallel speech data, where each input utterance is paired with a spoken form of its translation in the target language.

In order to reduce this dependency on parallel data, unsupervised S2ST systems (Wang et al., 2022b; Fu et al., 2023; Nachmani et al., 2023) that do not use any parallel data at all have been recently proposed. However, none of them are textless; these approaches either train non-end-to-end cascaded S2ST models (ASR-MT-TTS) in an unsupervised manner using unsupervised ASR (Liu et al., 2022b), unsupervised text-based MT (Liu et al., 2020) and unsupervised TTS (Liu et al., 2022a), or use text supervision during training (Nachmani et al., 2023). Thus, the crucial cross-lingual translation model is learned over text tokens, which limits their applicability to spoken languages.

Thus, existing S2ST work falls into two buckets: high-resource textless S2ST, and zero-resource text-based S2ST. Our work aims to bridge these two buckets by proposing a textless, low-resource S2ST model, which can be applied to spoken/unwritten languages without needing a lot of parallel speech translation data.

### 6.2 Textless and Unit-Based NLP

While we tackle textless S2ST, textless speech processing has studied in other tasks such as spoken language modeling (Borsos et al., 2022; Lakhotia et al., 2021; Hassid et al., 2024), emotion conversion (Kreuk et al., 2021), image-speech retrieval (Harwath et al., 2016; Peng & Harwath, 2022), spoken question answering (Lin et al., 2022) and speech evaluation (Chen et al., 2022; Besacier et al., 2023). Furthermore, progress in several other speech tasks like TTS (Wang et al., 2023) that involve both speech and text has been achieved by using powerful self-supervised units (semantic units like HuBERT (Hsu et al., 2021) and acoustic units like EnCodec (Défossez et al., 2022)).

## 7 Conclusion

We present the first textless low-resource speech-to-speech translation system, capable of learning from dozens of hours of parallel translation data, built by pretraining, finetuning, and backtranslating a language model over self-supervised speech unit sequences rather than text. We demonstrate its efficacy on 2 language pairs (English-German and English-Marathi) across 3 different domains. While our models achieve a decent translation performance, close to supervised toplines in some cases, they still underperform models trained on far more data or models that make use of text data, implying that several challenges still remain to make

these models highly performant. However, our approach holds great promise for modelling low-resource, primarily spoken languages. We hypothesize, based on similar findings for text machine translation, that scaling our approach to a larger unit LM pretrained on more data will improve performance and may unlock unsupervised textless S2ST akin to unsupervised text MT (Liu et al., 2020). Future work can investigate use of better S2U unit encoders for training better unit LMs, and training unit LMs on a larger set of languages.

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

## A  Datasets

| Module | Dataset | Duration | Lang |
|---|---|---|---|
| S2U Encoder: Pretraining | Librispeech | 960h | en |
| S2U Encoder: k-means Clustering | Librispeech, MLS
Shrutilipi | 48h, 48h
100h | en, de
mr |
| U2U Pretraining | Voxpopuli
Europarl-small
Europarl-mid
Shrutilipi | 529h, 248h
811h, 975h
2463h, 2918h
1000h | en, de
en, de
en, de
mr |
| U2U Finetuning (Toplines) | Europarl-ST
CVSS
Synth-EP-ST
Synth-Shr-ST | 83h,27h
91h,88h
83h,42h
76h,100h | en→de, de→en
en→de, de→en
en→mr, mr→en
en→mr, mr→en |
| U2U Finetuning (Low-Resource) | Europarl-ST
CVSS
Synth-EP-ST
Synth-Shr-ST | 10h,10h
10h,10h
30h,30h
30h,30h | en→de, de→en
en→de, de→en
en→mr, mr→en
en→mr, mr→en |
| U2U Backtranslation | Voxpopuli
Common Voice
Shrutilipi | 529h, 248h
294h, 89h
1000h | en, de
en, de
mr |
| U2S Vocoder | Voxpopuli
Shrutilipi | 529h, 248h
1000h | en, de
mr |
| Evaluation | Europarl-ST
CVSS
Synth-EP-ST
Synth-Shr-ST | 3h,6h
15h
9h
10h | en→de, de→en
de→en
mr→en
mr→en |

Table 4: Summary of datasets used to develop our system, with datasets used by base pretrained models colored red. Datasets in the U2U Finetune and U2U Evaluation sections are parallel translation datasets, and we report duration statistics for both translation directions separately, the duration being that of the source speech.

## B  Compute Details

We train all our models on 4 NVIDIA A40s (often using 2 GPUs with gradient accumulation of 2, or 1 GPU with gradient accumulation of 1, which is equivalent to 4 GPUs).

## C  S2U Encoder Ablations

To obtain the phoneme sequences for English and German, we use English and German phonemizers from the Montreal Forced Aligner[6]. For Marathi, we use a Kaldi-based ASR model trained on Shrutilipi data. First, we describe our ablations for English-German. We experiment with different base speech models (HuBERT vs. XLSR), layer indices, number of clusters (100 vs. 200) and types of clusterings (one clustering for both languages jointly v.s. separate clusterings) and choose the configuration that achieves the highest PNMI. We report PNMI results for some configurations in Figure 3.

---

[6]https://montreal-forced-aligner.readthedocs.io/en/latest/

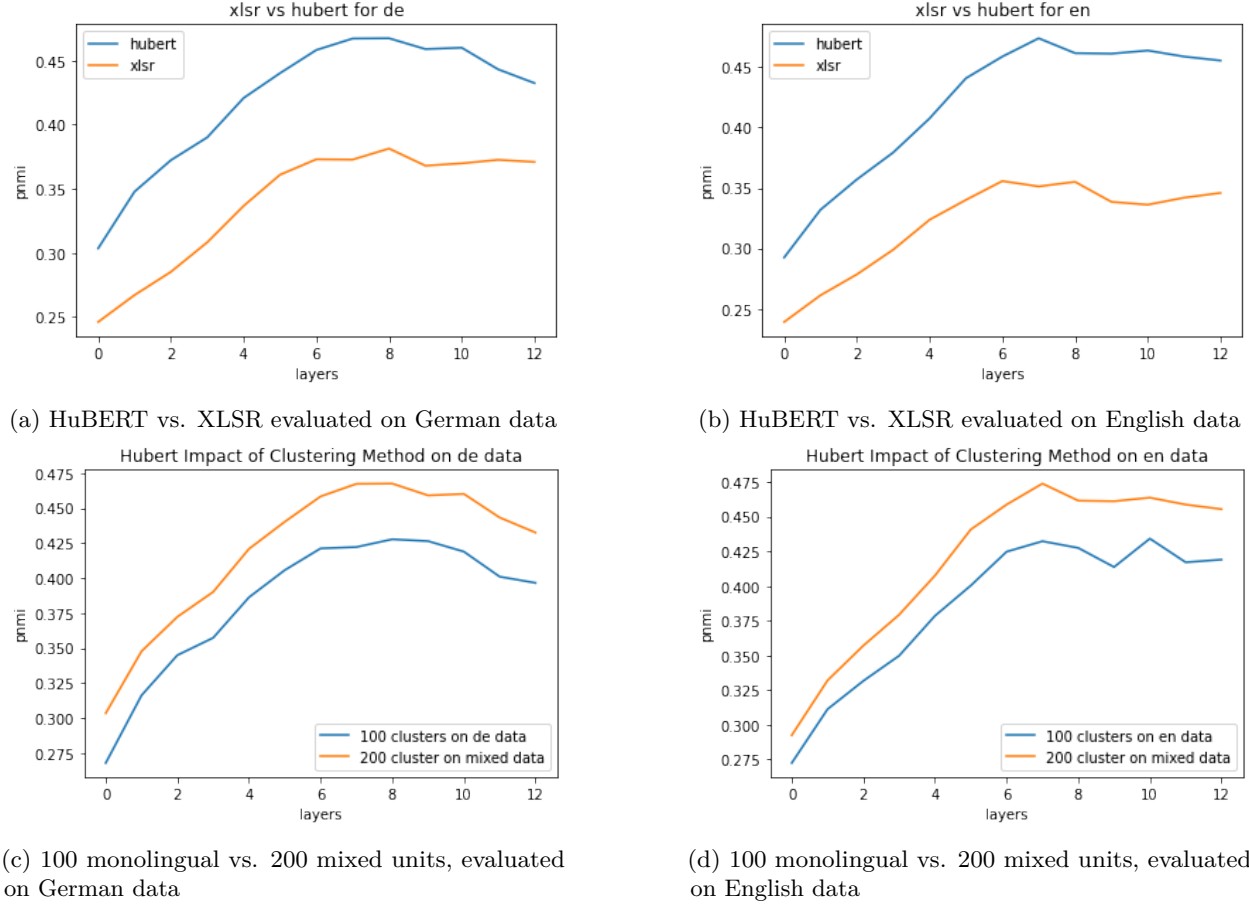

(a) HuBERT vs. XLSR evaluated on German data

(b) HuBERT vs. XLSR evaluated on English data

(c) 100 monolingual vs. 200 mixed units, evaluated on German data

(d) 100 monolingual vs. 200 mixed units, evaluated on English data

Figure 3: PNMI vs. layer index, comparing different clustering settings for English and German. Higher is better.

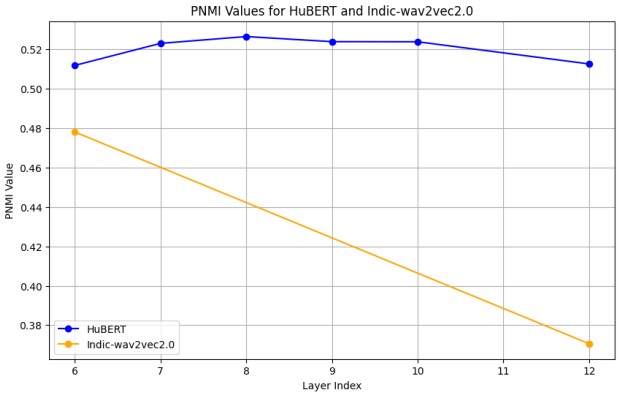

Figure 4: PNMI with HuBERT and Indic wav2vec2.0 evaluated on Shrutilipi, computed for different layer indices, for Marathi. Higher is better.

For Marathi, we experiment with different base speech models (HuBERT vs Indic-wav2vec2.0 (Javed et al., 2021)) and layer indices. We fix the number of clusters at 100. We choose the configuration that achieves the highest PNMI. We report PNMI results for some configurations in Figure 4.

| Method | en Voxpopuli | de Voxpopuli | en LJSpeech |
|---|---|---|---|
| Ground Truth | 4.89 | 8.44 | 3.80 |
| (Lee et al., 2022a) | 10.56 | - | 7.69 |
| Ours | 8.53 | 19.46 | 6.72 |

Table 5: S2U + U2S resynthesis performance; WER computed between resynthesized speech transcribed by ASR model and ground truth transcripts. Lower WER is better. We also include the ground-truth speech WER as a lower bound.

## D   S2U + U2S Resynthesis Evaluation

We compute the resynthesis WER as follows: (1) pass input speech to the S2U encoder and generate the unit sequence, (2) pass the generated unit sequence to our U2S vocoder to synthesize speech, (3) transcribe the synthesized speech using ASR (4) compute the Word Error Rate between the transcript and the ground truth transcript of the input speech. To account for the errors from ASR, we compute the WER between the ASR transcript of the input speech utterance ('ground-truth' speech) and the ground truth transcript as a lower bound. We use test sets from English and German Voxpopuli (Wang et al., 2021) and English LJSpeech (Ito & Johnson, 2017) with our synthetic single-speaker speech. Table 5 presents these results. We find that the resynthesis WERs are fairly good for English, and worse for German. Based on qualitative analysis of the German input speech (which is already single-speaker synthetic speech) and resynthesized speech (passed through S2U and U2S), we find that the input speech itself makes stress and pronunciation errors, driving up the Ground Truth WER, which further cascades into the model resynthesis WER. We still use this model because it is the best we could build with existing tools.

## E   Example Outputs

We present example outputs from our models. First, we showcase 10 cherry-picked examples, 2 examples from each evaluated language pair and domain in Table 6. Our best models, the post-backtranslation models (rows ⓙ and ⓠ in Tables 2 and 3) perform well on these examples. We present the ground-truth transcripts of the source and target utterances, the ASR transcript of the target utterance predicted by the pre-backtranslation finetuned models (rows ⓘ and ⓟ in Tables 2 and 3) and the ASR transcript of the target utterance predicted by our best models, the post-backtranslation models. We can observe that our post-backtranslation models are able to nearly perfectly translate these cherry-picked examples, which can be categorized into examples with (a) no mistakes (rows 1, 5, 7, 9), (b) valid replacements that largely preserve sentence meaning (rows 2, 4, 8) and (c) minor pronunciation errors (rows 6, 10). On the other hand, predictions from the finetuned model are overall worse, categorized into (a) no mistakes (row 1), (b) valid meaning-preserving replacements (row 2), (c) large meaning changes (row 3, 4, 7, 9, 10) and (d) incoherent output (row 5, 6, 8).

We also sample 5 randomly-picked examples, one from each setting to again compare our pre-backtranslation finetuned models and our best post-backtranslation models in Table 7. The examples show that the models are getting several of the words and semantics right, but often mistranslate certain words and make egregious grammatical and language modelling mistakes. We can see that our post-backtranslation model is overall better than the finetuned model for English-German in row (1), (2), worse in row (3), and performs similarly for rows (4) and (5).

| | Source Utterance | Target Utterance (Gold) | Prediction from fine-tuned model | Prediction from post-backtranslation model |
|---|---|---|---|---|
| | ***en→de (Europarl-ST)*** | | | |
| (1) | you can take initiatives | sie können initiativen er-greifen | sie können initiativen er-greifen | sie können initiativen er-greifen |
| (2) | madam president i supported this report | frau präsidentin ich habe diesen bericht unterstützt | frau präsidentin ich unter-stütze diesen bericht | frau präsidentin ich habe diesen bericht gestimmt |
| | ***de→en (Europarl-ST)*** | | | |
| (3) | ich denke da sind wir auf dem richtigen weg | i think we are on the right track here | i think we should be aware of this | i think we are on the right track |
| (4) | ich denke es ist klar dass die bürger und bürgerin-nen der europäischen union diese steuer wollen und ich denke dass es eine große verantwortung ist | i think it is clear that the citizens of the euro-pean union want this tax and i think we have a great responsibility here | i think that it is clear that the citizens of the european union want to do with these tasks and to do with the eu-ropean union what it wants to do | i think it is clear that the citizens of the european union want to be taxed and i think it is a major responsibility |
| | ***de→en (CVSS)*** | | | |
| (5) | stellst du die musik bitte auf zimmerlautstärke albert rief seine mutter | are you turning the volume down to room volume al-bert his mother screamed | are you turning the music albert towards its mountain rock | are you turning the volume down to room volume al-bert his mother screamed |
| (6) | los angeles liegt an der west-küste | los angeles is located on the west coast | loosen hot air line at the west coast | rose angeles is located on the west coast |
| | ***mr→en (s-Ep-ST)*** | | | |
| (7) | या कारणांमुळे मी या अह-वालाच्या बाजूने मत देऊ शकत नाही | for these reasons i cannot vote in favour of this report | for this reason i am in favour of the report | for these reasons i cannot vote in favour of this report |
| (8) | ते आधीच सुधारित केले गेले आहे परंतु आणखी काम करणे आवश्यक आहे | it has already been modified but more work needs to be done | it is improving barrowness improving but it must be forgotten | it has already made improvements but more work needs to be done |
| | ***mr→en (s-Shr-ST)*** | | | |
| (9) | पंचेचाळीस वर्षांवरच्या सर्वांनी लसीकरण अवश्य करुन घ्या | all those above forty five years must get vaccinated | more than forty five years of vaccination papers | all those above forty five years must get vaccinated |
| (10) | ते काल मुंबईत बातमीदारांशी बोलत होते | he was talking to reporters in mumbai yesterday | he was talking to reporters in mabay to day | he was talking to reporters in mumba yesterday |

Table 6: Cherry-picked examples picked for our best S2ST models (the post-backtranslation models), reporting predictions for both finetuned and post-backtranslation models. We manually annotate the differences between the gold utterance and the prediction from the post-backtranslation model, align them to the source utterance and underline the differences.

| | Source Utterance | Target Utterance (Gold) | Prediction from fine-tuned model | Prediction from post-backtranslation model |
|---|---|---|---|---|
| (1) | *en→de (Europarl-ST)* 
 goods and cargo have been delayed or not transported at all and businesses both large and small have been affected | waren und güterlieferungen wurden verschoben oder ganz gestoppt und sowohl kleine als auch große unternehmen sind betroffen | kosovo und konsum wurden zerstört oder wurden nicht erwähnt oder angemessen sein können | günstige und kunden wurden im vorle von kmos nicht erwähnt oder noch nicht erwähnt von allen unternehmen großen unternehmen |
| (2) | *de→en (Europarl-ST)* 
 wir sollten hier nicht mit zweierlei maß messen | we must not apply double standards here | we should not do so with these matters | we should not be here with the two sides |
| (3) | *de→en (CVSS)* 
 ihr schalldeckel trägt herabhängende quasten und ist mit einem pelikan bekrönt | their sounding board has loose hanging tassels and is crowned with a pelican | year study teacher however remaining costs and an ice and hobbies | child dictatorial territorial castes and is managed by a pellikov |
| (4) | *mr→en (s-Ep-ST)* 
 नैसर्गिक संसाधने आणि निसर्गांचे संरक्षण करण्यासाठी आपल्याला पर्यावरण संरक्षणाच्या क्षेत्रात संवादाची आवश्यकता आहे | we need dialogue in the field of environmental protection in order to conserve natural resources and nature | in order to protect natural resources and defense quality basis we need a clear signal of environmental protection | we need collectively in the area of protection resources for natural resources and jobs |
| (5) | *mr→en (s-Shr-ST)* 
 मुंबइ आणि उपनगरांमध्ये गेल्या काही दिवसांत जोरदार पाऊस झाल्यामुळ सात मुख्य तलावांच्या पाण्यात लक्षणीय वाढ झाल्यानं मुंबइला पुढील बारा महिने पाणी पुरवठा सुरळीतपणे होऊ शकणार आहे | heavy rains in mumbai and its suburbs in the last few days have significantly increased the water level in the seven main lakes ensuring smooth water supply to mumbai for the next twelve months | in the last few days ero people who have done in mumba mumbai soon reins have done in the last few days in the last few days mumbai | in mumba and opportunities of mumba and mumba who have received water in seventeen t h needs water in the last few days by the water in the mumbai |

Table 7: Randomly sampled examples comparing our finetuned and post-backtranslation models.

