# OpenReview forum: "Textless Low-Resource Speech-to-Speech Translation With Unit Language Models"
_TMLR — Rejected by TMLR_

### Review · Reviewer_Cryg · 2024-03-10

**Summary Of Contributions:**

Existing speech-to-speech translation models fall into two camps: textless models trained
with hundreds of hours of parallel speech data or unsupervised models that leverage text as
an intermediate step. This paper presents a framework for training
textless low-resource speech-to-speech translation systems that only need dozens of
hours of parallel speech data. The paper suggests a unit-to-unit translation model to perform the final task. Results are promising for this kind of translation systems.

**Audience:**

Yes

**Claims And Evidence:**

Yes

**Requested Changes:**

- While describing the back translation phase in your system, the paper mentions a single model M. This section is confusing and one might consult the previous work to properly understand the back translation properly. Please use a similar description in the related work and describe the technique with two models M1 and M2, where M1 translates from source to target language, and M2 translates from the target to source language. You may reuse some inference and train objectives from the previous work.

- Dataset abbreviations are not easy to follow. Should the prefix 'S' in the dataset S-EP-ST be capital or lower-case? There seems to be some sort of mismatch in the table and the paragraph description.

- How is the final translation performance of your system for shorter vs longer text? Such ablation would be insightful.

**Strengths And Weaknesses:**

Strengths:
The paper has successfully pre-trained a unit-to-unit translation system on generated speech units through a combination of three phases: a denoising phase, supervised fine-tuning using parallel resource and back translation. This suggests that future works could train massive unit-to-unit translation systems with bigger models. This is a promising direction for languages without written corpora.

Weaknesses:
- It would make the argument more stronger if you could run your text-less low-resource system on more language pairs without written text. Currently the paper only supports these types of languages with the only en -> mr translation task.

- It would make the comparison more insightful if some of the previous systems which are text-based or text-less with the same amount of data points you have used or even re-train their models on your datasets.

- The text-based zero-resource S2ST system in your baselines outperforms your system in high monolingual languages such as English and German which suggests that your proposed unit-based translation system should never prioritized on languages where they are high-resource monolingual written text.

---

> ### Author Response · Authors · 2024-04-03
> **Response to Reviewer Cryg**
>
> We thank the reviewer for their comments and suggestions. We respond to each concern below.
>
> **Q1: Experiments on more languages**
>
> **A:** Please refer to Q1 in our “General Response to Reviewers” for our response.
>
> **Q2: Text-based/textless systems trained on same amount of data as our models**
>
> **A:** Please refer to Q2 in our “General Response to Reviewers” for our response regarding text-based systems trained on the same amount of data. Textless systems are designed to be trained from scratch; as a result, they do not learn anything from just 20 hr of data, obtaining trivial ASR-BLEU scores of around 1.0 (as shown by row k and l of Table 2 in our paper); a pretraining step like our approach is necessary.
>
> **Q3: Text-based zero-resource S2ST system outperforms your model?**
>
> **A:** Yes, we agree that if one has access to high-resource monolingual written text, it is better to use a text-based S2ST topline model rather than our textless S2ST model. Our approach is designed for **textless S2ST** and is useful when at least one of the languages is textless i.e. has little to no text data resources, because then the text-based approach is impossible to build. We briefly discuss this in the paper in the first paragraph of Section 5.3, but we will add an explanation that it is better to use text-based models when there is ample text data available, and to use a textless low-resource approach like ours when there is little to no text data available.
>
> **Q4: Describe backtranslation phase better**
>
> **A:** We understand the reviewer’s concern about the section being confusing due to the denotation of a single model M. We described it as such because we are using **online backtranslation** in our paper, which sets the weights of the M1 model (which is used for forward inference) to the weights of the M2 model (which is used for computing the backward translation loss) immediately on every step; as a result, in every step, **M1=M2=M**, which is why we denoted it as M for ease of understanding. We realize that it may actually be more confusing to readers who are already familiar with backtranslation from previous papers (where M1 may not be equal to M2 e.g. in offline backtranslation), and will rewrite the section to use the more familiar, separated M1 and M2 models.
>
> **Q5: Dataset abbreviations are not easy to follow**
>
> **A:** The prefix ‘S’ in S-EP-ST is lowercase; it is rendered with the LaTeX \textsc{} command which is visually appealing but makes it look like a shorter uppercase S. We apologize that this was confusing and will change it. We understand that overall the dataset abbreviations may have been hard to follow; we will include a global table of all abbreviations and their definitions in the paper so that it is easy to follow them.
>
> **Q6: Translation performance for shorter vs longer text?**
>
> **A:** We agree that this is an interesting analysis and we have performed it. For each test dataset (Europarl-ST, CVSS, synth-EP-ST and synth-Shruti-ST), we first compute the character lengths of every target example in the test set and compute the 33rd and 66th percentiles of the length distribution. We consider all examples with a length shorter than the 33rd percentile to be ‘short’, ones in between the two to be ‘medium’, and ones longer than the 66th percentile to be ‘long’. We then compute ASR-BLEUs separately over each of these subsets for the models in row j of Table 2 and row q of Table 3, and show them here:
> |Model and Dataset | |ASR-BLEU | | |
> |:----|:----|:----|:----|:----|
> | |short|medium|long|all|
> |Row j (Europarl-ST de -> en)|10.1|10.6|9.5|10.0|
> |Row j (Europarl-ST en -> de)|9.6|9.0|7.7|8.3|
> |Row j (CVSS de -> en)|6.5|8.3|7.7|7.7|
> |Row q (synth-EP-ST mr -> en)|10.9|10.1|8.0|9.2|
> |Row q (synth-Shruti-ST mr -> en)|10.9|13.0|8.0|10.0|
>
> The results show that the model does better on short/medium utterances than long utterances, which is in line with intuition. It is also reassuring that the performance of the long utterances is within just 1-2 BLEU points of the overall performance.

---

> > ### Comment · Reviewer_Cryg · 2024-04-04
> > **Response to Authors**
> >
> > - Thank you for the clarification on the online back translation and the extra ablation about performance for short vs. long speech.
> >
> > - I am a bit confused about the computational costs of your approach as you mentioned it takes around 2 months to run experiments for an extra language pair. It would be great to describe the computational costs with respect to the number of training examples and the type of models you are running.

---

> ### Author Response · Authors · 2024-04-05
> **Response 2 to Reviewer Cryg**
>
> Thank you for looking over our response. The computational cost we mentioned is broadly the development time for the English-German model we trained in the paper i.e. mBART-50 model pretrained on speech unit sequences on Voxpopuli and Europarl data on 4 NVIDIA A40 GPUs. The exact sizes of datasets, the number of steps and other important hyperparameters are there in our paper in Section 4.2 and Table 4. The time we reported included the time it takes to find good hyperparameters, train a model, wait for our jobs to get scheduled on our compute cluster, etc. Overall, the development cycle from choosing a new language pair to obtaining a model built for that language pair is around 2 months, primarily because we need to pretrain a fresh model for each language pair.
>
> We hope this clarifies things, but please feel free to ask any follow-up questions you have about it and we would be happy to respond.

---

### Review · Reviewer_cqXW · 2024-03-20

**Summary Of Contributions:**

This study introduces a speech-to-speech translation model that does not require corresponding text data. Additionally, the authors claim that their models are trained on a limited amount of parallel training data, which sets them apart from previous approaches that rely on text data or very large-scale parallel speech data.

**Audience:**

Yes

**Claims And Evidence:**

Yes

**Requested Changes:**

1. It is recommended that the authors incorporate a Mean Opinion Score (MOS) test to assess the speech quality. This would provide a more comprehensive evaluation and enhance the credibility of the results.
2. The research necessitates a comprehensive investigation into the semantic or linguistic characteristics of the speech units.
3. Further discussion on the model's generalizability to different languages and its salability to larger datasets would be beneficial.

**Strengths And Weaknesses:**

strength:
1. The authors' description of their methods, including the framework and implementations, is presented in a highly clear manner.
2. Some results reported in Table 2 and Table 6 are promising and encouraging.

weakness:
1. While the model utilizes a small-scale parallel training dataset, it necessitates a large-scale pretraining dataset, such as the pretrained speech2unit model and the pretrained text2text model.
2. The ablation experiments presented in Table 2 highlight the advantages of utilizing backtranslation. However, it is important to note that the performance of the ablation models significantly lags behind the majority of the baseline models.
3. The absence of speech samples makes it challenging to evaluate the quality of the generated speech. It is recommended that the authors incorporate a Mean Opinion Score (MOS) test to assess the speech quality. This would provide a more comprehensive evaluation and enhance the credibility of the results.
4. The research necessitates a comprehensive investigation into the semantic or linguistic characteristics of the speech units.
5. Further discussion on the model's generalizability to different languages and its salability to larger datasets would be beneficial. Addressing these aspects would provide insights into the model's potential for real-world deployment and its adaptability to diverse linguistic contexts.

---

> ### Author Response · Authors · 2024-04-03
> **Response to Reviewer cqXW**
>
> We thank the reviewer for their comments and suggestions and are glad that they found the presentation clear and appreciated our results as well. We address their concerns below.
>
> **Q1: Needs a large-scale pretraining dataset**
>
> **A:** Yes, we certainly need a large pretraining dataset, since we perform a crucial self-supervised pretraining step which requires several hundreds of hours of monolingual data. It is exactly due to the **high-quality representations** learned by this step that we are able to train textless S2ST models that only require a few tens of hours of parallel data. This is in contrast to previous textless S2ST papers that used hundreds of hours of parallel data to train non-trivial systems. The main point is that the large-scale pretraining dataset is monolingual speech data, which is cheap to scrape and collect at scale, while parallel training data requires expensive human annotations. Therefore, our approach is *trading expensive parallel data for cheap monolingual data, which is a **strength**!
>
> **Q2: Ablation experiments lag behind baseline models**
>
> **A:** We note that the ablation experiments (rows k-n in Table 2) are exactly meant to show that they are **worse** than our best approach (row j), showing that each component of our pipeline is useful. It is not surprising that the ablation experiments lag behind the other models.
>
> **Q3: Mean Opinion Score (MOS) tests to assess speech quality**
>
> **A:** MOS tests are typically useful when one wishes to compare the speech naturalness or compare non-content aspects of speech, which are difficult to do without human evaluators. In this paper, we are primarily interested in evaluating the **semantic content** of the generated speech as we are concerned with the correctness of the translation. For this purpose, the ASR-BLEU metric is usually considered a sufficient evaluation criterion. See past papers like https://arxiv.org/pdf/2305.07455.pdf , https://arxiv.org/abs/2211.04508 which only use the ASR-BLEU metric. Other papers that do use MOS e.g. https://proceedings.mlr.press/v162/jia22b.html , https://arxiv.org/abs/2308.01831, use it to evaluate speech naturalness, not speech content. While improving the naturalness of the model’s outputs is a clear direction for future work, we are first focusing on the semantic correctness of the output. We will clarify in the text that the evaluation reflects the semantic content of the speech, rather than any other aspects of the speech.
>
> **Q4: Investigation into the semantic characteristics of speech units**
>
> **A:** We do perform a detailed comparison study of the semantic characteristics of the speech units in Section 4.1 and Appendix C, which shows that we use speech units that have the **highest measured semantic content** (as measured by the phoneme PNMI metric described in the paper). We note that HuBERT and HuBERT-like models (that we use) have been successfully used in several previous papers for textless language models (GSLM https://arxiv.org/abs/2102.01192, TWIST https://arxiv.org/abs/2305.13009) and unit-based S2ST (https://arxiv.org/pdf/2308.01831.pdf, https://arxiv.org/pdf/2112.08352.pdf) as well, which provides more support for its use in our paper.
>
> **Q5: Model generalizability to more languages and larger datasets**
>
> **A:** Please refer to Q1 in our “General Response” for our response.

---

> > ### Comment · Reviewer_cqXW · 2024-04-25
> >
> > I would like to express my sincere appreciation for the highly informative responses provided. The majority of my concerns have been thoroughly addressed, and I wholeheartedly recognize the value of the additional results that demonstrate the training of other models on an equivalent dataset. However, I still believe that conducting further investigation into the features of the speech units is of utmost importance, given the significant reliance of this method on the pretrained speech2unit model. Specifically, conducting a thorough analysis of the linguistic and acoustic features of the speech units, coupled with a comprehensive evaluation of the generated speech quality (not just semantic evaluations), would undoubtedly yield invaluable insights. Considering the inherent nature of this model as a speech-to-speech generation model, such investigations would be particularly pertinent and contribute significantly to the overall understanding and advancement of this work.

---

### Review · Reviewer_eehf · 2024-03-25

**Summary Of Contributions:**

This submission describes an approach for building speech-to-speech translation models with limited quantities of supervised data. It combines existing approaches such as pre-training, unit-to-unit translation, fine-tuning and back-translation. Three languages were considered: English, German and Marathi. Of those only Marathi could be viewed as a limited resource language (~80 million speakers). Text data excluding pre-training came from Europarl, Common Voice and All India radio broadcasts. All speech data was synthetic and generated by a single speaker text-to-speech system.

**Audience:**

Yes

**Claims And Evidence:**

No

**Requested Changes:**

Please see the issues mentioned above. In particular, please focus on the list of contributions if you have any, experiments that better feature your approach, experiments that provide more valuable contrasts (e.g. your UTUT but with standard text-based MT units (one of T2T baselines)).

Given that there are real rather than synthetic datasets for S2ST please provide a rationale for not using them at all.

Unclear who is "Seamless Communication"?

**Strengths And Weaknesses:**

The key strength is that this submission describes a very sensible low-resource approach to speech-to-speech translation, which combines all the standard approaches such as pre-training, fine-tuning and back-translation. The use of a previously proposed unit-to-unit translation approach makes this direction towards low-resource languages/configurations quite appealing.

There are a number of weaknesses unfortunately.

Perhaps the major one is that the submission makes it really hard to understand the true contribution of this work. The pre-train, fine-tune and back-translate strategy is hardly new although the authors make it sound so. The unit-to-unit translation is not new although the authors also make it sound so. Perhaps consider producing a list of contributions if you have any and if not then clearly stating so.

Another major weakness is the experimental methodology. Looking at (ASR-)BLEU scores I am not certain that an S2ST model with a BLEU<10 could be considered a successful approach for limited resource S2ST. Perhaps consider showing examples to argue that such systems are nevertheless useful (although some info is available in the appendix it is never discussed). Looking at the cascaded system I see that it dates back to 2019 and perhaps is significantly underestimated given a huge progress in ASR and MT areas. I was also hoping to see some T2T baseline but could not find any. Although you have looked at 3 languages across several datasets I was expecting to see a much broader and deeper investigation than you presented. A substantial number of your numbers comes from other people considering different approaches and architectures which provides a very limited light on your specific approach. Most of your ablation study shows configurations with BLEU<5 which does not tell much given how low is the BLEU score.

---

> ### Author Response · Authors · 2024-04-03
> **Response to Reviewer eehf**
>
> We thank the reviewer for their comments and suggestions. We would like to address your concerns below:
>
> **Q1: Difficult to understand the true contribution; the approach is hardly new, but the authors make it sound so. Please produce a list of contributions.**
>
> **A:** Our main contribution is addressing **textless low-resource S2ST**, a gap in existing literature that focuses on either high-resource textless S2ST or low-resource text-based S2ST. While the reviewer is correct (and we acknowledge in our paper) that each step of the framework is previously studied, their adaptation to **new domains** (e.g. backtranslation over spoken units instead of text), is a non-trivial research contribution.
>
> Specifically, our **contributions** are:
> 1. Tackling the task of textless low-resource S2ST and proposing a unit-to-unit model for it
> 2. Adapting the pretrain-finetune-backtranslate approach for unit-based S2ST, showcasing backtranslation over spoken units for the first time
> 3. Releasing open-source textless models for English-German and English-Marathi
>
> We fully acknowledge the source of each framework step by **citing relevant past papers**, and do not claim any of the steps to be a fully novel idea; we cite mBART-50 (https://direct.mit.edu/tacl/article/doi/10.1162/tacl_a_00343/96484/Multilingual-Denoising-Pre-training-for-Neural) for the pretrain step and https://arxiv.org/abs/1711.00043 for the backtranslate step. Adapting this for unit-based S2ST was not straightforward; we had to ablate the right units, use curriculum-learning pretraining, use supervised replay during backtranslation, etc. We believe that we are one of the first to use **unit-to-unit translation for textless S2ST**; while unit-based speech LMs are not new (e.g. GSLM, AudioLM), we believe we are one of the first to use it for S2ST (the only other approach we find https://arxiv.org/abs/2308.01831 is contemporaneous). That being said, if the reviewer cites specific papers that do unit-to-unit S2ST, we would be happy to cite them and fix our claim.
>
> We hope that this set of contributions sufficiently satisfies the reviewer’s request and also the acceptance criteria of TMLR (https://jmlr.org/tmlr/acceptance-criteria.html), which emphasizes technical correctness over a subjective notion of significance.
>
> **Q2: Is BLEU<10 a successful approach for limited resource S2ST? Consider showing examples.**
>
> **A:** We agree that an ASR-BLEU of around 10 is modest; we only refer to these results as ‘non-trivial’ in the paper. However, under the **immense data constraints** (textless + low-resource), this is an important result and challenging to achieve. Our results compare decently well to the textless topline models (rows f-h) in Table 2; despite being trained with far more parallel data, they have similarly low ASR-BLEUs < 16. This underscores the **inherent difficulty** of training fully textless S2ST models. Futher, the samples in Appendix E (Table 7) show that despite large grammatical and faithfulness issues, the model captures critical information from the source language, which is useful. We will add a discussion regarding this to the paper.
>
> **Q3: Cascaded system is from 2019 and is likely underestimated**
>
> **A:** The topline from 2019 was only included as a rough reference rather than a SoTA model; we apologize for the confusing wording. To rectify this issue, we are updating the paper with 2 new toplines. Please refer to Q1 in our “General Response to Reviewers” for a new **cascaded low-resource topline**. Secondly, we created a more recent **high-resource cascaded S2T topline** by cascading Whisper ASR Medium (https://arxiv.org/abs/2212.04356) with NLLB-200 1.3B (https://arxiv.org/abs/2207.04672) for Europarl-ST, getting BLEU scores of 25.8 for en-de and 26.5 for de-en. Just NLLB-200 1.3B (i.e. a T2T topline) gets BLEU scores of 26.2 for en-de and 28.4 for de-en.
>
> **Q4: Needs broader and deeper investigation. A T2T baseline is missing.**
>
> **A:** Please refer to Q1 and Q2 in our “General Response to Reviewers” for our responses. Apart from a text-based topline, could the reviewer expand upon any other specific experiments they would like to see in the paper?
>
> **Q5: Why use synthetic datasets?**
>
> **A:** Modeling real speech data with speech unit sequences poses challenges related to the unitizer quality that are **orthogonal** to our research question (investigating the potential for textless low-resource S2ST), such as inherent unit sequence noise and ambiguity. Thus, for simplicity, we use single-speaker synthesized speech data to train and evaluate our models, which is similar to early S2ST work (https://arxiv.org/abs/1904.06037). We discuss this in the paper in the paragraph below Figure 1.
>
> **Q6: Who is ‘Seamless Communication’?**
>
> **A:** We cite them in the paper: https://arxiv.org/abs/2308.11596 . The official citation has the first author listed as ‘Seamless Communication’ which makes the citation appear as ‘Seamless Communication’.

---

### Author Response · Authors · 2024-04-02
**General Response to Reviewers**

We thank all the reviewers for their insightful comments, questions and suggestions. We are glad that all reviewers found the approach meaningful and reasonable. We respond to concerns raised by multiple reviewers in this general response, and other questions and concerns in individual responses to each reviewer.

**Q1. by Reviewers Cryg and cqXW: Experiments on more languages and datasets**

**A.** We currently perform analysis on 3 translation directions (English to and from German, Marathi to English), where Marathi is a real low-resource language, and across 3 different domains (European Parliament, Common Voice and All India Radio). We acknowledge that adding more textless language pairs will strengthen our paper’s main argument. Unfortunately, due to the absence of pre-existing open-source textless multilingual generative speech models, adding new language pairs incurs **significant computational costs** on our side, primarily due to the pretraining part of our pipeline that needs to be done from scratch for each language pair, taking about 0.5 months to optimize important hyperparameters and then about 1.5 months to train a final pretrained model (on 4 NVIDIA A40s), i.e. almost 2 months per new language pair. We have initially piloted our ideas on one language pair (English-German) and then later expanded our study to English-Marathi, and found **our results generalized fairly consistently**, despite many differences between two languages. Given this, we anticipate our method can generalize to other languages, which can be done by future work.

**Q2. by Reviewers Cryg and eehf: Text-based systems trained on same amount of data as our models**

**A.** We thank the reviewers for this insightful comment. We have run the requested experiment for text-based systems on Europarl English-German translation. To recall, to build our unit S2ST model, we pretrain an mBART-50 model on English and German monolingual speech data and finetune on 20 hours of English-German S2ST data. We now describe a **low-resource text-based topline** that is as close as possible to this setting. We finetune the pretrained mBART-50 text model, on the text transcripts corresponding to the same 20 hours of parallel S2ST data. We use the same hyperparameters as our model. This serves as a text-to-text (T2T) translation topline. We can create a speech-to-text (S2T) translation topline by cascading an ASR system with this finetuned mBART model. We can similarly create a speech-to-speech (S2ST) translation topline by cascading an ASR system, the finetuned mBART, and a TTS system. We use the same ASR models used for computing ASR-BLEU (details in Section 5.1), and the same TTS models used for generating our single-speaker data (details in Section 3.3). **Our results are as follows** (we denote our finetuned mBART as ft-mBART):
|Approach|de->en ASR-BLEU|en->de ASR-BLEU|
|:----|:----|:----|
|T2T topline: input text --(ft-mBART)--> output text translation|27.7|26|
|S2T topline: input speech --(ASR)--> ASR input text --(ft-mBART)--> output text translation|25.2|23|
|S2ST topline: input speech --(ASR)--> ASR input text --(ft-mBART)--> output text translation --(TTS)--> output speech translation|23.7|21.3|
|Rows from Table 2 of paper| | |
|a) Cascaded ASR-MT (Iranzo-Sánchez et al., 2019)|21.3|22.4|
|b) E2E S2T (Wang et al., 2021)|17.5|-|
|c) E2E S2T w/ Voxpop-Aligned (Wang et al., 2021)|18.8|-|
|f) Bilingual S2S (Duquenne et al., 2022)|16.3|10.1|
|g) Multilingual UTUT (Kim et al., 2023)|15.8|9.8|
|h) Pretrain + Fully Finetune (Ours)|12|13.4|
|i) Pretrain + Finetune on 20hrs (Ours)|7.8|6.8|
|j) Pretrain + Finetune on 20 hrs + Backtranslate|10|8.3|

With 20 hrs of text data, these low-resource text-based toplines outperform other text-based (rows a-c) and textless (rows f-h) toplines, as well as our low-resource models. This is not surprising since learning over text sequences is dramatically easier than learning over speech unit sequences, as text is more semantically grounded and less noisy than speech units; we make this point in the paper when comparing S2T toplines to S2ST toplines as well. Overall, these new toplines **highlight the task difficulty** of training low-resource textless S2ST systems as compared to text S2ST systems.

---

### Author Response · Authors · 2024-04-17
**Thank you for reviewing. Please let us know if you have additional questions!**

We thank all the reviewers for their helpful suggestions and their insightful questions. Please let us know if you have any additional questions and suggestions before the end of the discussion period, so that we can work towards resolving them! Thank you.

---

### Decision · Action_Editor_wSEx · 2024-05-12

**Recommendation:** Reject

**Comment:**

While reviewers acknowledge the efforts that the author made both in the submission and rebuttal period to demonstrate the usefulness of this method, there are still important concerns such as lack of investigation into the speech units and lack of adequate language pairs for low-resource scenarios. Suggest the authors to improve according to these points and consider submitting the improved version to other venues.

**Audience:**

With limited interest to the audience in speech-to-speech translation

**Claims And Evidence:**

While some claims are well supported by evidence, reviewers still believe that conducting further investigation into the features of the speech units is of critical importance to the overall understanding and advancement of this work. Meanwhile, reviewers think that more language pairs are necessary to demonstrate the usefulness of proposed approaches.